# Spatiotemporal analysis of bubonic plague in Pernambuco, northeast of Brazil: Case study in the municipality of Exu

Diego Leandro Reis da Silva Fernandes[1], Elainne Christine de Souza Gomes[2], Matheus Filgueira Bezerra [1], Ricardo José de Paula Souza e Guimarães [3‡], Alzira Maria Paiva de Almeida [1‡*]

1 Department of Microbiology, Institute Aggeu Magalhães–Fiocruz PE, Recife, Pernambuco, Brazil,
2 Department of Parasitology, Institute Aggeu Magalhães–Fiocruz PE, Recife, Pernambuco, Brazil,
3 Geoprocessing Laboratory, Evandro Chagas Institute/SVS/MS, Brazil

ʘ These authors contributed equally to this work.
‡ RJPSG and AMPA also contributed equally to this work.
* aalmeida@cpqam.fiocruz.br

**Data Availability Statement:** In this reviewed version we added to the supplemental files our data bank on patients clinical and epidemiological features, as well as a case-by-case sheet with all

## Abstract

Along with other countries in America, plague reached Brazil through the sea routes during the third pandemic. A brief ports phase was followed by an urban phase that took place in smaller inland cities and finally, it attained the rural area and established several foci where the ecological conditions were suitable for its continued existence. However, the geographic dispersion of plague in Brazil is still poorly studied. To better understand the disease dynamics, we accessed satellite-based data to trace the spatial occurrence and distribution of human plague cases in Pernambuco, Northeastern Brazil and using the municipality of Exu as study case area. Along with the satellite data, a historical survey using the Plague Control Program files was applied to characterize the spatial and temporal dispersion of cases in the period of 1945–1976. Kernel density estimation, spatial and temporal clusters with statistical significance and maximum entropy modeling were used for spatial data analysis, by means of the spatial analysis software packages. The use of geostatistical tools allowed evidencing the shift of the infection from the urban to the wild-sylvatic areas and the reemergence of cases after a period of quiescence, independent of the reintroduction from other plague areas.

## Introduction

Plague is a focal zoonosis, affecting primarily rodents and eventually, humans and other mammals. The infection spectrum is wide and the main presentations are the bubonic, pneumonic and septicemic forms; the contamination occurs mainly through flea bites, inhalation of aerosols or contact with infected secretions or tissues [1]. It is caused by *Yersinia pesti*s, a Gram-negative bacillus that belong to the Enterobacteriaceae family [2] and is categorized in the Biohazard Class 3 and Bioterrorism Agents Group A [3]. This zoonosis is one of the oldest and

confirmed plague cases in the State of Pernambuco, including dates, locations and the coordinates for each municipality affected.

**Funding:** The author(s) received no specific funding for this work.

**Competing interests:** The authors have declared that no competing interests exist.

most feared diseases of mankind and remains a threat still nowadays. Over the centuries, this infection wiped out millions of lives, impacted the people way of life, having a huge influence in science and arts over the ages and it still represents the iconic perception of a pandemics until nowadays [4].

Currently, the global incidence of human plague is the lowest reported by the WHO in 30 years. During the 2013–2018 period, the areas that have reported human cases were limited to sub-Saharan Africa, Asia, and North and South America, and most cases are reported by Madagascar, followed by the Democratic Republic of the Congo [5].

However, it is not uncommon the sudden reappearance of cases after several decades of epidemiological silence in natural plague foci causing fear, panic and loss of life and in the economics. In 2003 the plague reappeared in Algeria after >50 years of quiescence [6]. In 1994, the city of Surat, India, experienced a pneumonic plague outbreak that caused panic, population evasion and a severe impact on local economy [7]. Other episodes occurred in 2009 in the city of Ziketan, China, and in 2019 in the Mongolian border with China and Russia which resulted in the closure of the borders between these countries [5].

Along with other countries in America, plague only reached Brazil during the third pandemic, in the year of 1899. Transported by steamships, the disease caused its first outbreaks in port cities. However, due to inland transportation of goods, plague quickly reached Brazilian countryside, establishing several natural foci where the ecological conditions were congenial for its persistence [8]. The Brazilian plague foci are scattered throughout a large area ranging from the Northeastern State of Ceará to the Southeastern State of Minas Gerais and another separated area at the State of Rio de Janeiro [9–11].

The activities of the infection in these foci are independent in time and space [9]; after several successive outbreaks of varied sizes until the decade of 1980, cases in Brazil decreased and the last confirmed human case was in 2005 [12, 13]. Despite its current quiescent state, we must remain vigilant and maintain rigorous epidemiological surveillance. Of note, attempts to eradicate plague by some countries (USA, USSR) were unsuccessful and therefore this was discontinued and disapproved [14, 15].

The availability of new technological resources allowed the development of new studies on the plague activities, which is crucial for improving the understanding of the disease dynamics and to establish effective monitoring and surveillance strategies, capable of recognizing eventual issues that may precede spillovers to human populations. This study aimed to better understand the plague dynamics in Pernambuco, Northeastern Brazil by analyzing the spatial occurrence and distribution of cases and using the municipality of Exu as study case area by performing a historical survey of human cases and characterizing the spatial and temporal dispersion of cases.

## Methods

### Study area

The study was carried out in the state of Pernambuco, Northeast Brazil, and the municipality of Exu was used for the case study of plague. This municipality lies in the mesoregion *Sertão*, has an area of 1,336,788 km$^2$, an estimated population in 2019 of 31,825 inhabitants, Municipal Human Development Index (2010) of 0.576 and a warm and dry climate with scarcity and irregular rainfall (Biome *Caatinga*). Situated in the ecological complex of Chapada do Araripe, 600–700 m in altitude, about 200 km long and 30 km wide, is limited to the municipalities of Bodocó to the west, Granito to the south, Moreilândia to the east and to the north with Crato in the state of Ceará [16, 17].

## Data collection

Data on the occurrence of human plague cases in the state of Pernambuco were obtained from the forms named *Comunicado Sobre Ocorrência de Peste Humana* (Notification on the Occurrence of Human Plague) from the Plague Control Program; CONCEPAS system (http://cyp.fiocruz.br/index?services); and activity records from the Plague Laboratory, available in the National Reference Service in Plague of the Aggeu Magalhães Institute, Fiocruz PE.

Information on the early occurrences of plague in the state is sparse and incomplete, therefore for the 1902 to 1944 period only data relating the occurrence of the first case per municipality were obtained. For the 1945 to 1976 period, collected data included the date and place of the occurrence, patient's gender, age, clinical features, and classification of the case (suspected, positive or negative). All data were compiled and organized into a database (DB) using Excel software.

For geospatial analyses, the vector data obtained were: municipal limits of 1970 and 2010 from the Brazilian Institute of Geography and Statistics (IBGE) (https://www.ibge.gov.br/geociencias/organizacao-do-territorio/15774-malhas.html?=&t=downloads). The drainage (hydrography) from the Mineral Resources Research Company (CPRM) (https://www.cprm.gov.br/en/Hydrology-83). The Digital Elevation Model (DEM) data was obtained from the Shuttle Radar Topography Mission (SRTM) refined for the Brazilian territory from the original resolution of 3 arc seconds to 1 arc second using a geostatistical approach (http://www.dsr.inpe.br/topodata/) using the script (https://code.earthengine.google.com/ccf3b9ff46eb845e1b88f68550e9a22a) on the Google Earth Engine (GEE) platform. All geospatial data were obtained from free access and use platforms.

## Data analysis

The DB was separated into two groups: (1) DBM (Database by Municipality), containing both the years of the first plague case by municipality during 1902–1966, and all cases recorded by municipality for the 1945–1976 period; (2) DBE (Exu Database), which was separated into two sub-groups corresponding to the epidemic periods of 1945–1954 and 1961–1976 and a quiescence period of 1955–1960. The localities of the DBE cases (suspected, positive or negative) in the study period (1945–1976) were georeferenced in loco, with a GPS (Global Positioning System), model eTrex Vista Cx, Garmin (Kansas City, USA), configured in the UTM (Universal Transverse Mercator) projection system, Datum WGS-84. To georeference, a landmark (house, church or gate) was standardized for each of the localities. The GPS data was transferred to GPS TrackMaker Pro 4.9.603 (Geo Studio Technology, Belo Horizonte, Brazil) and the geographic coordinates were organized and stored in the shape file format, that was used with the DB to create a spatial database (SDB).

Descriptive epidemiology was used to analyze the distribution of cases by gender, clinical features, age, location (urban or rural area) and period of the occurrence. Unfortunately, due to the lack of standardization of the records over time, some of the clinical variables were not available for all patients.

The spatial analyzes performed in these groups were: (1) map of spatial and temporal distribution to spatially visualize the location of the disease and the number of cases in the municipalities of Pernambuco and in the localities (*sitio*, *fazenda*, *povoado*) of Exu (choropleth maps); (2) Kernel density estimation (KDE) to identify the location of clusters for case occurrences. For KDE, the following parameters were used: quadratic function, density calculation and adaptive radius on both banks; (3) spatial scanning map (Scan) to identify spatial and temporal clusters with statistical significance. The Scan used the Poisson model (Retrospective Space-Time analysis scanning for clusters with high rates using the Discrete Poisson model) based on

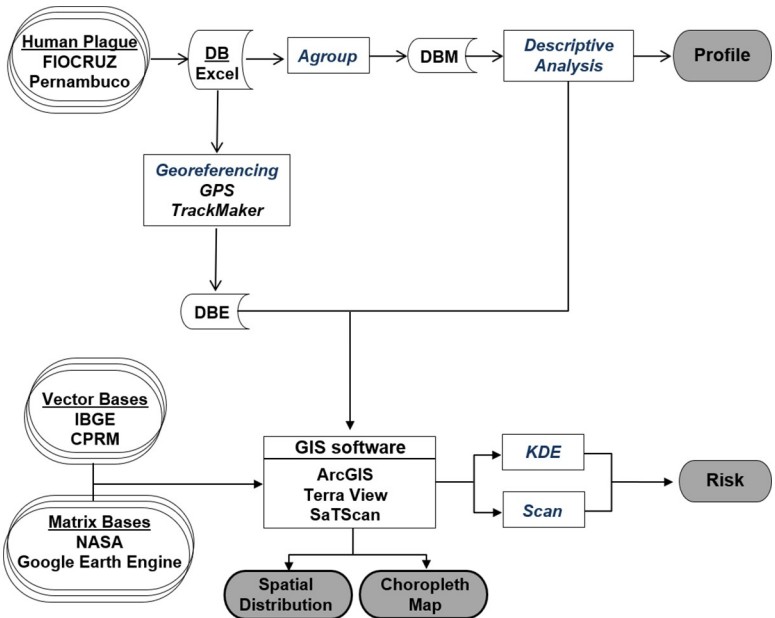

**Fig 1. Workflow for collection, storage and analysis of spatiotemporal data.**

the resident population in Pernambuco for the analysis in the DBM and the Exponential model using the DEM attribute obtained by Google Earth Engine (GEE) to search for spatial and/or temporal clusters of exceptionally short or long survival in DBE (Retrospective Space-Time analysis scanning for clusters with short or long survival using the Exponential model).

Data processing, interpretation, visualization and analysis were performed using ArcGIS (http://www.arcgis.com/), SatScan (https://www.satscan.org/) and TerraView (http://www.dpi.inpe.br/terralib5/wiki/doku.php). Fig 1 illustrates the methodology for collecting, storing and analyzing spatiotemporal data. The surface and boundaries of the municipality of Exu during the study period are different from the present, because some small rural communes called *Povoados* (Tabocas, Viração, Timorante) were emancipated and became new urban areas named *Vilas* (Villages).

## Results and discussion

### Entry and dissemination of the human plague in Pernambuco, Northeast Brazil

In 1902, just three years after the plague entered Brazil, the state of Pernambuco was affected by the disease [11]. According to the data collected in this study, from the introduction until the last case (1982), 56 municipalities (30.4%) of the extant 184, registered plague cases (Fig 2). Since its arrival, strict sanitary control measures undertaken eliminated the infection from the port city (Fig 2A), however, they failed to prevent its spread to the countryside [8, 11]. During the period of 1913–1918, the disease spread to the *Agreste* mesoregion, reaching eight clustered municipalities (Fig 2B). In 1919, the plague spread further throughout the state, reaching two new mesoregions, one municipality in the *Zona da Mata* or Forest Zone and four in the *Sertão* (Fig 2C). From 1920–1936, 16 new municipalities were attaint for the first time including an only one municipality in São Francisco mesoregion (Fig 2D).

The Fig 2E and 2F presents the two largest epidemic periods for the state. In the first period, 1945–1954, only one new municipality was affected, in the year 1948 (Fig 2E), in the second,

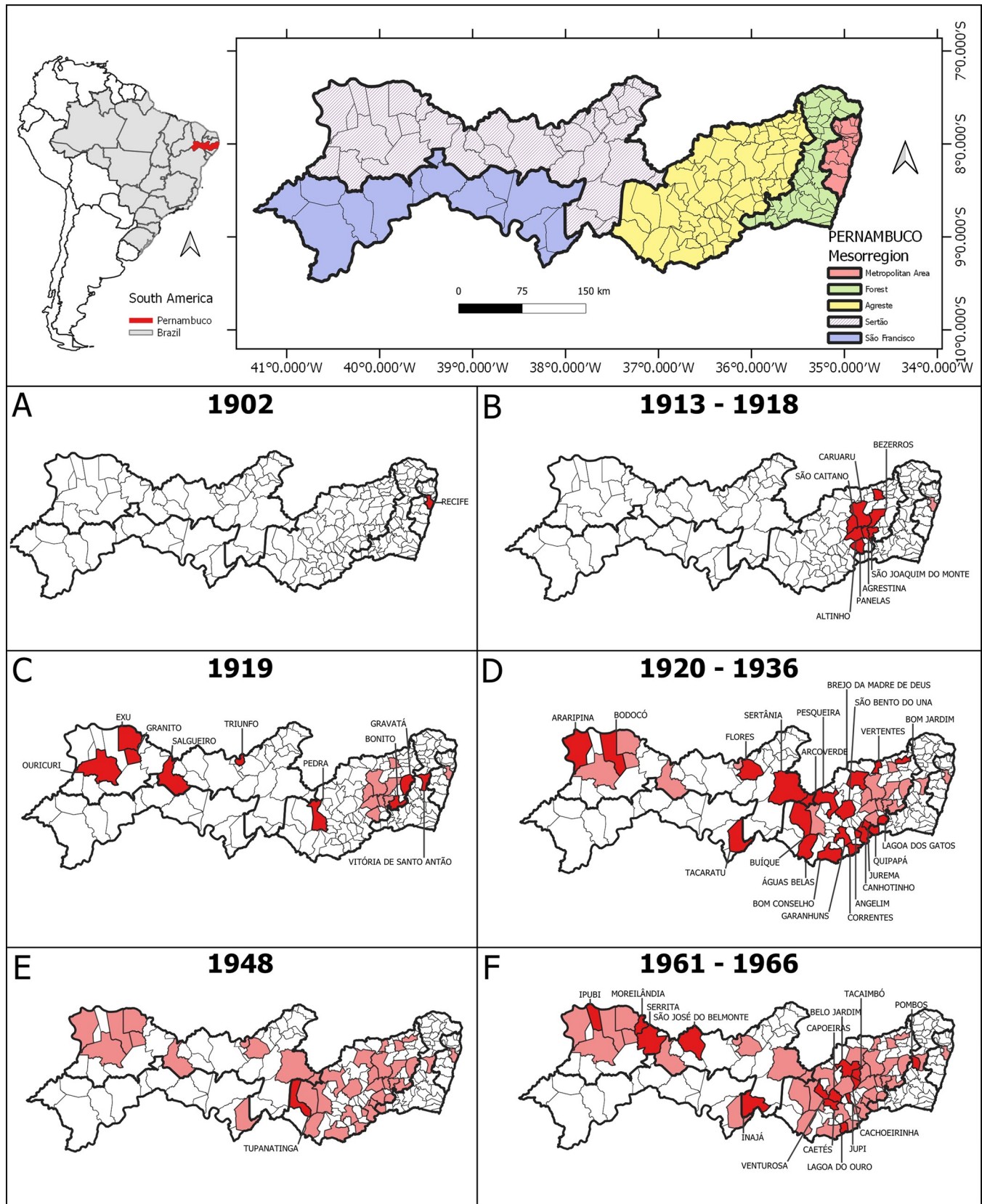

**Fig 2. Emergence and dissemination of human plague cases by municipality in the mesoregions of the state of Pernambuco, Northeast Brazil, 1902–1966.** On top: Localization of Pernambuco and Brazil in South America and the mesoregions. A-F: Spatial and temporal distribution of the human plague cases, showing the municipalities affected for the first time (dark red) and those previously affected (light red). Shapefiles of Pernambuco and counties limits were obtained from IBGE (public-domain access).

1961–1966, which was the period with the highest number of cases, 14 new municipalities registered plague cases (Fig 2F). It is important to note that after 1966, no new municipality registered occurrence of plague case until 1982, the date of the last notification in the state.

## Establishment of the plague in Pernambuco, Northeast Brazil

According to the epidemiological records, there were 954 plague-suspected cases notified in Pernambuco during the period from 1945 to 1976. Out of these, 525 (55.0%) were considered positive based on the epidemiological-clinical criterion for the classification of cases at the time [9, 11].

The 954 plague-suspected and the 525 positive cases originated from 57 (31%) and 37 (20%), respectively, of the extant 184 municipalities in the state of Pernambuco (Fig 3A). Exu (red) concentrates the largest number of positive cases (267), followed by Bodocó (45 cases),

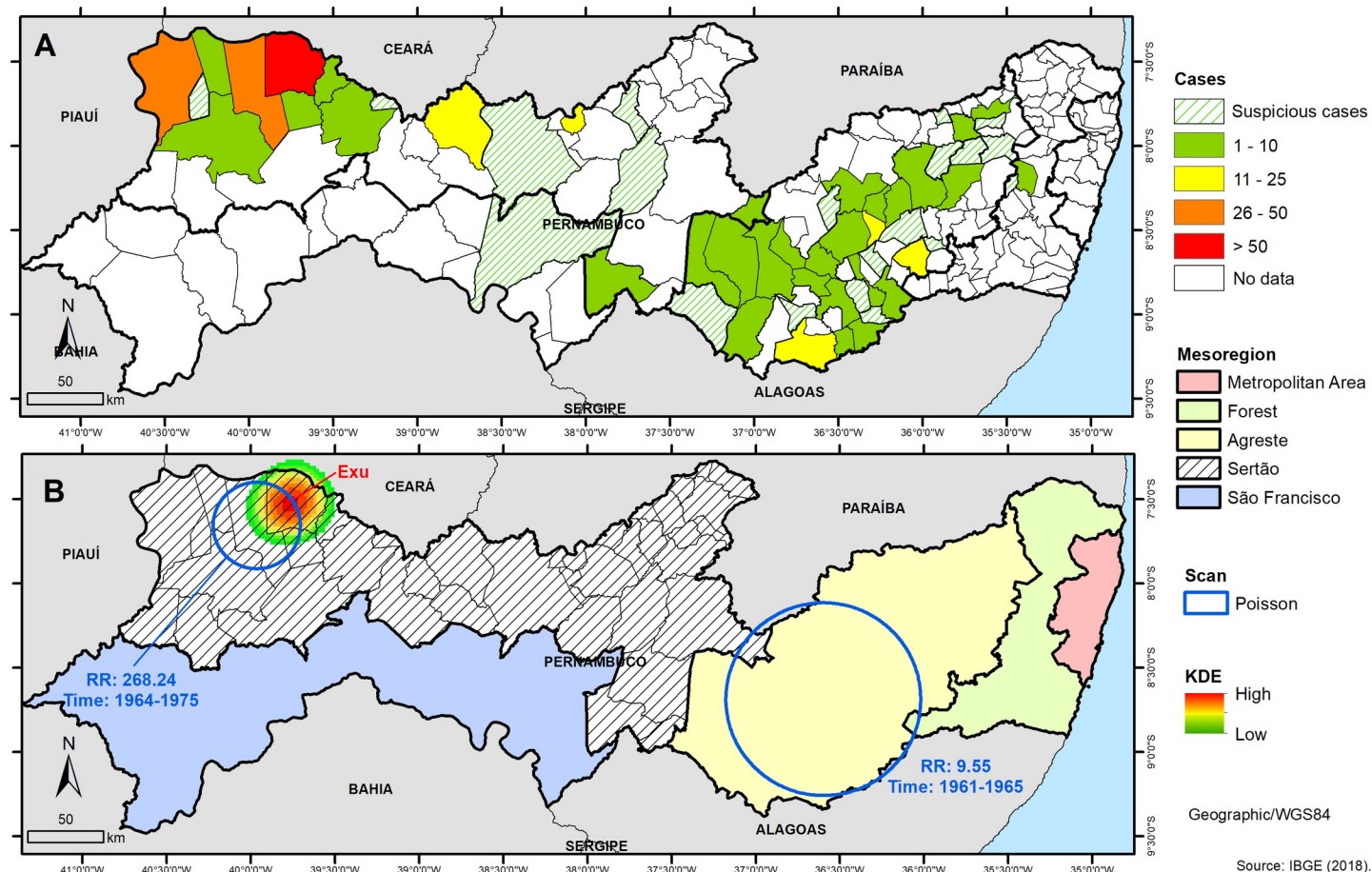

**Fig 3. Spatial distribution and risk analysis of human plague cases in Pernambuco, Northeast Brazil between the years 1945–1976 with the identification of the mesoregions.** (A) Spatial distribution of human plague by number of occurrences per municipality; (B) Identification of risk areas for the occurrence of the disease by application of KDE and Scan in cases of human plague in Pernambuco. Shapefiles of Pernambuco and counties limits were obtained from IBGE (public-domain access).

Araripina (37 cases) and other five municipalities, all located in the ecological complex of Chapada do Araripe, in the *Sertão* mesoregion. In this same mesoregion, the ecological complex of Triunfo—São José do Belmonte registered 17 and 12 positive cases (yellow), respectively. This mesoregion, despite not concentrating the largest number of plague afflicted municipalities, is doubtless of great epidemiological importance, since it concentrates the largest number of plague cases in Pernambuco during the studied period (398 cases—75,8% of the total positive cases).

In the *Agreste* mesoregion, the highest number of positive cases was registered in the municipalities of Bom Conselho (20), Cachoeirinha (13) and Panelas (11) and other 31 municipalities registered one to 10 plague-positive cases. These municipalities are located in the Planalto da Borborema, another focal plague area involved in the 1961–1976 epidemics in the state of Pernambuco.

It is noteworthy that in the period of the study, when the plague was already established in Pernambuco, no new cases were reported in the metropolitan region or in the mesoregion of *São Francisco*, although in this later at least one case had been registered in the 1920–1936 period as seen in Fig 2D. In the *Forest* mesoregion, only one municipality registered three positive cases (green). The absence of cases in these areas could be related to their ecological and geographical conditions, such as altitude, vegetation and the temperature unsuitable for the persistence of the infection. In contrast, in the *Agreste* and *Sertão* mesoregions, the plague encountered the congenial ecotopes to maintain the wild cycle and establish permanent foci in the ecological complexes of Chapada do Araripe and Planalto da Borborema [9, 10].

The Kernel density analysis (KDE) of the number of cases reported in Pernambuco revealed that the municipality of Exu from the Chapada do Araripe focus is at higher risk for the occurrence of plague (Fig 3B). Exu appeared at the epicenter of the Kernel patch that radiates in decreasing intensity as it moves away from the Chapada slope towards the plains and the neighboring municipalities of Moreilândia, Bodocó, Granito and Crato. The Fig 3B shows as well the presence of two statistically significant spatiotemporal clusters obtained by the Scan analysis, one that encompasses the municipality of Exu with a relative risk (RR) 268.24 times higher for the occurrence of plague in comparison with the other municipalities of Pernambuco in the period 1964–1975 and another cluster located in the *Agreste* mesoregion with RR = 9.55 in the period 1961–1965.

These preliminary findings led us to perform a more detailed analysis using the Municipality of Exu as a model to better understand the dynamics of plague's transmission and maintenance in Brazil.

## Plague in Exu

From the 525 positive cases in the state of Pernambuco, 267 (50.9%) originated in the municipality of Exu. All cases reported in Exu in the analyzed period were the primary bubonic form, characterized by the presence of the bubo, which was generally single, extremely painful, accompanied by high fever and torpor. The temperature (recorded in 43 patients) ranged from 37˚ to 41˚C (mean 38.7˚C) and the buboes (recorded in 42 patients) located in the following anatomical sites: inguinal-crural (36), axillary (4), cervical (2). There was no record of pneumonic plague, but 14 patients had septicemia confirmed by a positive *Y. pestis* blood culture and this septicemia was probably evolution from the primary bubonic infection [18, 19].

Although data on response to treatment and patient follow-up were unavailable for several cases, those that could be accessed had satisfactory response to the standard treatment by Sulfadiazine and Streptomycin and most cases recovered; the few deaths were attributed to lack or late treatment as stated by Karimi et al. [18]. According to Freitas [19] with the use of sulfa-

antibiotics therapy replacing the use of serum in treatment from 1943 on, deaths declined significantly.

Regarding the age at diagnosis, patients were grouped in: early childhood (0 to 6 years = 44), childhood (7 to 12 years = 39), adolescents (13 to 18 = 20), adults (19 to 59 = 38) and elderly (≥60 = 3). Among the patients, 220 were men (61.1%) and 140 women (38.9%). Regarding seasonality, from 1961 to 1976, although cases were recorded in all months of the year but February and April, most cases occurred from July to November, with 8 to 17 cases per month. A positive association was observed between the plague cases and increased rodent reservoir and flea vector populations and higher number of naturally infected rodent and fleas in the fields in the municipality of Exu [9, 18, 20].

Analysis of the spatial distribution of the cases reveals that Exu city (urban area) concentrates the largest number of cases and is surrounded by smaller spots in neighbor rural sites. The spatial contiguity between them favors the exchange between the commensal and wild rodent hosts (Fig 4). Of note, once the infection disseminated to the rural zone, most affected areas (localities x cases) were concentrated close to the Chapada slope, while rather dispersed in the plateau (brown). Indeed, according to Baltazard [9], all the plague cases from the municipalities of Exu and Bodocó occurred along the green and fertile slopes with numerous islets of dense brushwood, permeated by springs issues from the Chapada. The few cases in the contiguous plains were limited to a 30–50 km zone neighboring the Chapada slopes. This is a congenial ecosystem for the coexistence of the wild rodent reservoirs of the bacteria, the flea vectors and the humans [21].

In order to study the dissemination of plague cases in the municipality of Exu over time, three distinct periods were stratified: (1) ten years of activity (1945–1954); (2) quiescence for six years (1955–1960) and (3) 15 years of activity (1961–1975). Based on this stratification, it was possible to visualize the dissemination of the plague from the urban to the rural areas (Fig 5A and 5B).

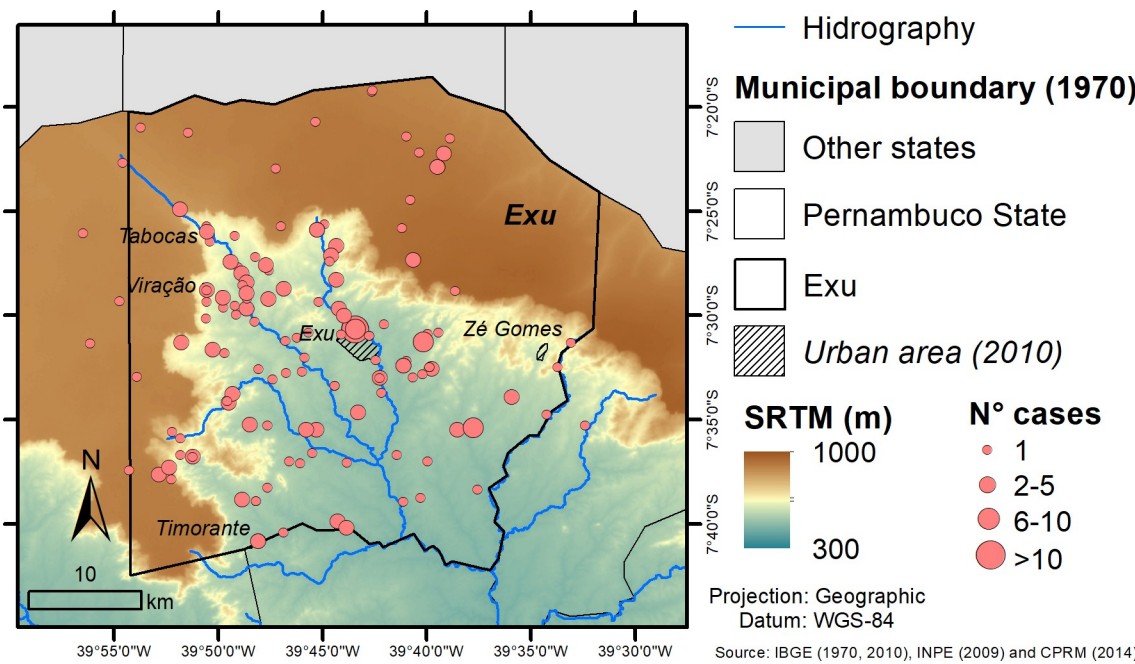

**Fig 4. Spatial distribution of human plague cases in Exu, Pernambuco, Northeast Brazil, in the period 1945–1976 considering the number of cases per locality and the relationship with altimetry.** Shapefile of Exu was obtained from IBGE; DEM from SRTM (http://www.dsr.inpe.br/topodata/); and Hydrography from CPRM. Images were used for illustrative purposes only.

In the period of 1945–1954, from the 74 cases in the municipality, 32 (43%) were urban and 42 (57%) were rural, spread in 33 localities. In Exu city the most numerous cases were recorded in 1945 (9) and 1946 (19), followed by 1948 (2), 1951 (1) and 1952 (1). This period can be considered the end of the urban phase (Fig 5A) when the infection raged into the city houses through the population of the commensal rats (*Rattus rattus*) and was likely transmitted by the rat fleas (*Xenopsylla cheopis*) or by *Pulex irritans*, the so called "human flea" [22] and even by *Ctenocephalides felis*, the cat fleas. Karimi et al. [20] reported the occurrence of infected free-living *X. cheopis* and *P. irritans* in the soil of the houses or in the bedding of the deceased patients from Exu. Further evidence of the potential role of *C. felis* and *P. irritans* in plague transmission was provided elsewhere [23–25]. No cases were registered in the city during the following 11 years (1952–1963), reappearing only in 1964, three years after the re-emergence and the outburst of plague cases in the rural areas.

In contrast to the first epidemic period (1945–1954), there were only 9 cases in the city (4.7%) in the second period (1961–1975), while in the rural area there were 184 cases (95.3%) (Fig 5B). This change in the occurrences pattern could be due to the control measures employed to eliminate rats and fleas from the city houses based on the use of poisons and insecticidal (DDT and BHC) spraying [11, 19]. On the other hand, the continued pressure of spraying insecticides up to three times a year for many years, led to the insecticidal resistance among the *Xenopsylla* and *Pulex* flea populations, which may have contributed to the upsurge of the plague in the 1960s [9, 20].

Fig 5C shows that at the beginning of the first period (1945–1954), the cases were concentrated in the city of Exu and surroundings as well as in the small rural communes and surroundings such as the *povoados* Tabocas, Viração and Timorante (green) and at the end of the period (red) the cases occurred further away from the city. In the beginning of the second period (Fig 5D), the cases were concentrated in the rural area around the *povoados* Viração and Tabocas (green), and at the end of the period (red) they spread across the plain towards the *povoado* Timorante and to the south, to the neighboring municipality of Bodocó.

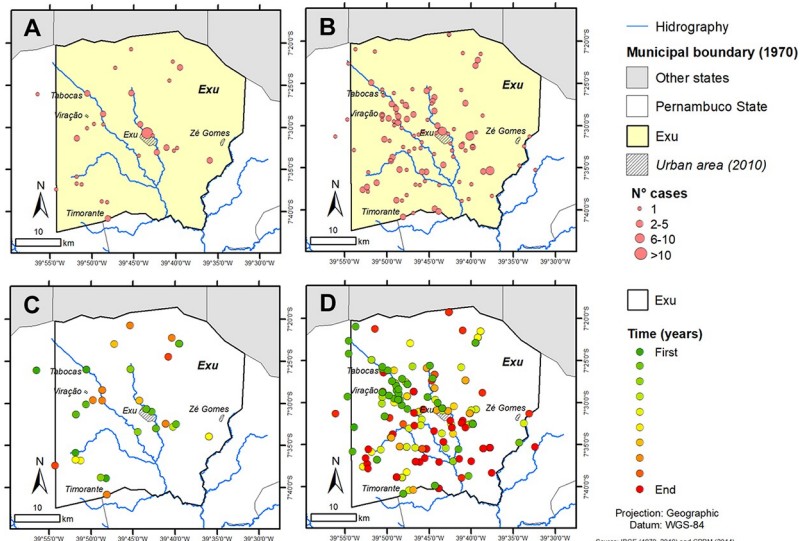

**Fig 5. Spatiotemporal distribution of human plague cases in Exu, Pernambuco, Northeast Brazil.** (A) Showing the concentration of human plague cases in the urban area of Exu in the period of 1945–1955; (B) Dispersion of cases to the rural areas during the second epidemic period 1961–1975; and the timeline shift within each period stratified by color scale: dark green, cases at the beginning of each period and dark red, those at the end of each period (C) distribution of cases by locality 1945–1954 and (D) 1961–1975. Shapefile of Exu was obtained from IBGE; and Hydrography from CPRM. Image was used for illustrative purposes only.

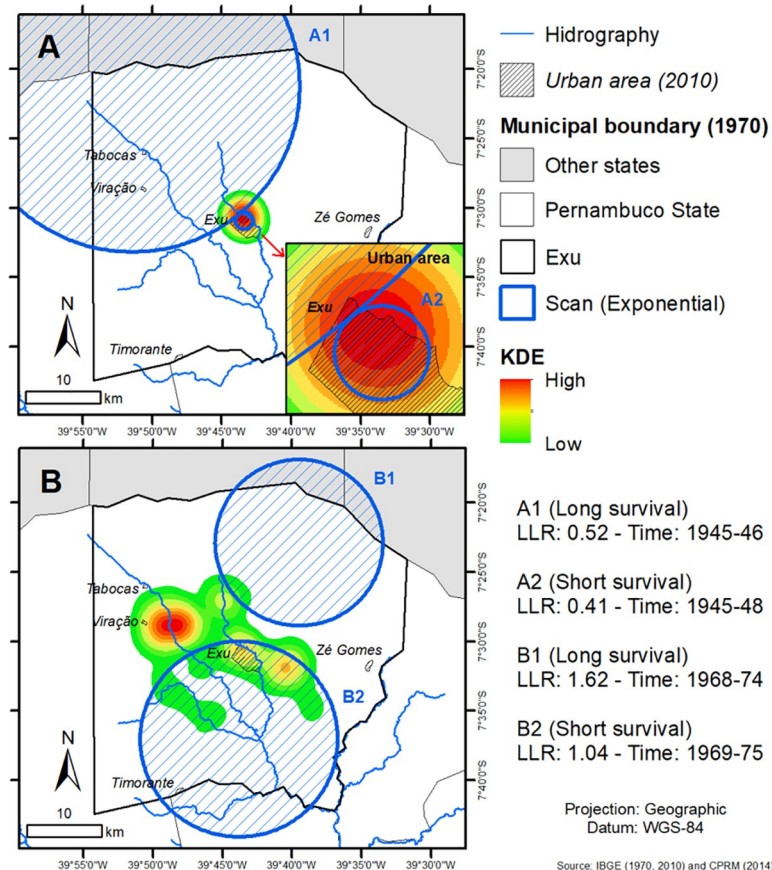

**Fig 6.** Application of KDE and Scan in the human plague cases in Exu, Pernambuco, Northeast Brazil in the period 1945–1955 (A) and 1961–1975 (B). Retrospective Space-Time analysis scanning for clusters with short or long survival using the Exponential model. Log Likelihood Ratio (LLR). Shapefile of Exu was obtained from IBGE; and Hydrography from CPRM. Image was used for illustrative purposes only.

Such findings demonstrate that the first epidemics started in the city urban area and disseminated to the rural areas mainly where the human detector was more numerous. In the second epidemics, which counted the largest number of cases (193–72.3% of cases), the plague was already disseminated practically throughout all the municipality territory. In this period the plague was well established among the wild fauna [21]. The rodent *Necromys lasiurus* is the most common wild rodent, the most frequently infected by plague and carrying infected fleas (*Polygenis bolhsi jordani* and *P. tripus*) which played likely the more important role in the spread of plague and the human cases [9, 18, 20].

The analysis of risk areas by KDE in the municipality of Exu confirmed the dynamics of the plague dispersion. In the first epidemic period (1945–1954) cases predominated in the city— urban phase, represented by the risk spot (Fig 6A). It is also possible to note the presence of two significant spatiotemporal clusters obtained by the Scan analysis, one with a long survival rate located in the urban center of Exu with an LLR of 0.41 in the period from 1945–1948 and another with a short survival rate located in the northwest part of the municipality (rural area) with LLR = 0.52 in the period from 1945–1946. Such findings highlight the higher risk and duration for the occurrence of cases in the urban zone in that period, certainly associated with commensal rats *R. rattus* and the rat fleas *X. cheopis* [20].

In the second epidemic period (1961–1975), the risk spot for the occurrence of cases (KDE) is concentrated in the rural area, mainly in the *povoados* Viração, Tabocas and Zé Gomes (Fig 6B), where the number of human detectors was higher. There was also a marked presence of two significant spatiotemporal clusters obtained by the Scan analysis, one with a long survival rate located in the northern part of the municipality (rural area) with LLR = 1.62 in the period 1968–1974 and the other with short survival rate located in the part south of the municipality (enrolling the urban center of Exu) with LLR = 1.04 in the period 1969–1975. These results demonstrate the higher prevalence of the plague in the rural areas during this second epidemic period in Exu.

These results enlighten the dynamics of the epidemization and epizootization of the plague in Exu. However, the absence of cases in the quiescence period (1955–1960) could be misleading considering that there was no investigation of the circulation of the bacillus in the nature and the only indicator of the infection activity was the human cases and it can be assumed that the occurrences could be under-reported [9].

Globally, the mechanisms of persistence of the infection in the plague foci in different regions of the world during the interepizootic periods are not yet fully understood [4, 14]. It was hypothesized that during the quiescence period the infection could remain in the nature in an enzootic form as a low-level circulation of the rodent-flea-rodent cycle with more time in the fleas' organism or in the rodents as a chronic form of the plague. It is also postulated that the bacteria would endure inside the burrows of certain rodent species, where the microclimate would allow its survival in debris from dead animals, in the soil contaminated with flea feces and in soil parasites [9, 14, 21, 26].

## Conclusion

During the studied period it was demonstrated the transition of the infection from the urban to the rural areas in Exu, Pernambuco state, Northeast of Brazil and the reemergence of cases after a quiescence period without reintroduction from other foci. After six years of quiescence, the plague reappeared in 1961 in the rural area, in a *sitio* 16 km far from the city of Exu. This case would be served as an alert and the epidemics from 1964 onward could had been avoided or constrained. However, there was not a surveillance system or a predictive model at that time. Therefore, the case was not properly considered and consequently, in the following 15 years until 1975, the plague spread throughout the municipality territory and reached places that had not been affected in the previous epidemic periods.

At the beginning of the first epidemic period analyzed (1945–1954), the infection was still raging mainly in the urban area maintained by the commensal rats and their fleas and then moved to the rural area from which it reemerged after 6 years of silence, reappearing in the city only three years later while it was already largely active on the rural areas. Unexpectedly after a long period of continuous activity and at a time of larger expansion when it reached several municipalities in the Chapada do Araripe, the plague suddenly disappeared in this focus since 1975. This disappearance was confirmed not only by the absence of human cases but also by the absence of the plague bacillus among the rodent hosts and flea vectors and a decreasing in positive sentinel/indicator animals through the bacteriological and serological surveillance activities [12, 27]. The causes for this disappearance are not known. Purportedly it could be attributed to the rarefaction of rodent populations decimated by successive epizootics and which were unable to recover due to climatic changes [9].

This change of pattern was surprising and contrary to what was expected, as predicted by Baltazard [9], which would be a gradual reduction of the cases and the retreat of the infected area with persistence in Exu until the total halt of the infection activities in the focus.

However, this silence must not be interpreted as extinction of the focus, because at any time, by an unknown mechanism the infection can reactivate. Due to its cyclical characteristics—alternating periods of activity and quiescence, depending on a series of complex factors—the plague can reemerge, causing new epizootics and reaching the human populations [28, 29].

Since it is so widespread in wildlife rodent reservoirs and considering the particularities of the focal areas, the eradication of the plague is a momentarily unattainable objective and not even recommended in the face of the failure of the eradication attempts carried out by some countries [15, 30]. It is therefore essential to maintain the monitoring and control of this zoonosis in order to avoid future spillovers for the human populations.

## Supporting information

**S1 Data.**
(XLSX)

## Author Contributions

**Conceptualization:** Diego Leandro Reis da Silva Fernandes, Elainne Christine de Souza Gomes, Ricardo José de Paula Souza e Guimarães, Alzira Maria Paiva de Almeida.

**Data curation:** Diego Leandro Reis da Silva Fernandes.

**Formal analysis:** Diego Leandro Reis da Silva Fernandes, Elainne Christine de Souza Gomes, Matheus Filgueira Bezerra, Ricardo José de Paula Souza e Guimarães.

**Funding acquisition:** Alzira Maria Paiva de Almeida.

**Project administration:** Alzira Maria Paiva de Almeida.

**Software:** Ricardo José de Paula Souza e Guimarães.

**Supervision:** Elainne Christine de Souza Gomes.

**Writing – original draft:** Diego Leandro Reis da Silva Fernandes, Elainne Christine de Souza Gomes, Matheus Filgueira Bezerra, Ricardo José de Paula Souza e Guimarães.

**Writing – review & editing:** Diego Leandro Reis da Silva Fernandes, Elainne Christine de Souza Gomes, Matheus Filgueira Bezerra, Ricardo José de Paula Souza e Guimarães, Alzira Maria Paiva de Almeida.

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
