## [Decision Letter · Decision Letter 0]

11 Jan 2021

PONE-D-20-35938

Spatiotemporal analysis of bubonic plague in Pernambuco, Northeast of Brazil: case study in the Municipality of Exu.

PLOS ONE

Dear Dr. Almeida,

Thank you very much for submitting your manuscript, "Spatiotemporal analysis of bubonic plague in Pernambuco, Northeast of Brazil: case study in the Municipality of Exu" (PONE-D-20-35938), for consideration at PLOS ONE. As with all papers reviewed by the journal, your manuscript was reviewed by members of the editorial board and by several independent reviewers. In light of the reviews (below this email), we would like to invite the resubmission of a significantly-revised version that takes into account the reviewers' comments.

We look forward to receiving your revised manuscript.

Kind regards,

Abdallah M. Samy, PhD

Academic Editor

PLOS ONE

**Additional Editor Comments:**

Please address carefully all comments below from Reviewer #1; we have to unassign any other reviewers to avoid any further delay on this manuscript. 

**Journal Requirements:**

2. We note that Figure 1 in your submission contains copyrighted images. All PLOS content is published under the Creative Commons Attribution License (CC BY 4.0), which means that the manuscript, images, and Supporting Information files will be freely available online, and any third party is permitted to access, download, copy, distribute, and use these materials in any way, even commercially, with proper attribution. For more information, see our copyright guidelines: http://journals.plos.org/plosone/s/licenses-and-copyright.

(1) You may seek permission from the original copyright holder of Figure 1 to publish the content specifically under the CC BY 4.0 license.

3. We note that Figures 2-6 in your submission contain map images which may be copyrighted. All PLOS content is published under the Creative Commons Attribution License (CC BY 4.0), which means that the manuscript, images, and Supporting Information files will be freely available online, and any third party is permitted to access, download, copy, distribute, and use these materials in any way, even commercially, with proper attribution. For these reasons, we cannot publish previously copyrighted maps or satellite images created using proprietary data, such as Google software (Google Maps, Street View, and Earth). For more information, see our copyright guidelines: http://journals.plos.org/plosone/s/licenses-and-copyright.

(1) You may seek permission from the original copyright holder of Figure(s) [#] to publish the content specifically under the CC BY 4.0 license. 

**Reviewers' comments:**

Reviewer's Responses to Questions

**Comments to the Author**

1. Is the manuscript technically sound, and do the data support the conclusions?

Reviewer #1: Yes

2. Has the statistical analysis been performed appropriately and rigorously? 

Reviewer #1: No

3. Have the authors made all data underlying the findings in their manuscript fully available?

Reviewer #1: No

4. Is the manuscript presented in an intelligible fashion and written in standard English?

Reviewer #1: Yes

5. Review Comments to the Author

Reviewer #1: I feel the paper has the potential to be published in this journal. First, the authors have did a good job to edit the historical data of plague in South America. I appreciate this very much. The data is very valuable with detailed time and location information.

However, the author had a poor data analysis. The paper is too descriptive to be accepted at the current form. There are available approaches to estimate the transmission rates using historical data (e.g. Xu et al. 2014. Wet climate and transportation routes accelerate spread of human plague. Proceedings of the Royal Society B-Biological Sciences. 281: 1780), and then analyze the potential factors affecting the transmission rate, and identify the key factors.

6. PLOS authors have the option to publish the peer review history of their article (what does this mean?). If published, this will include your full peer review and any attached files.

Reviewer #1: No

---

## [Author Response · Author response to Decision Letter 0]

3 Feb 2021

ADDITIONAL EDITOR COMMENTS/ JOURNAL REQUIREMENTS:

Response: We followed the PLOS ONE's style requirements.

2. We note that Figure 1 in your submission contains copyrighted images. All PLOS content is published under the Creative Commons Attribution License (CC BY 4.0), which means that the manuscript, images, and Supporting Information files will be freely available online, and any third party is permitted to access, download, copy, distribute, and use these materials in any way, even commercially, with proper attribution. For more information, see our copyright guidelines: http://journals.plos.org/plosone/s/licenses-and-copyright.

We require you to either (1) present written permission from the copyright holder to publish these figures specifically under the CC BY 4.0 license, or (2) remove the figures from your submission.

Response: In this new version of the manuscript, we provide a modified version of Figure 1 that does not contain any images or logos under copyright protection.

3. We note that Figures 2-6 in your submission contain map images which may be copyrighted. All PLOS content is published under the Creative Commons Attribution License (CC BY 4.0), which means that the manuscript, images, and Supporting Information files will be freely available online, and any third party is permitted to access, download, copy, distribute, and use these materials in any way, even commercially, with proper attribution. For these reasons, we cannot publish previously copyrighted maps or satellite images created using proprietary data, such as Google software (Google Maps, Street View, and Earth). For more information, see our copyright guidelines: http://journals.plos.org/plosone/s/licenses-and-copyright.

We require you to either (1) present written permission from the copyright holder to publish these figures specifically under the CC BY 4.0 license, or (2) remove the figures from your submission.

Response: We thank the editorial team for bringing this issue into attention. In this reviewed version, we added into the Methods session the links to access the databases used to build figures 2-6. It is, however, important to highlight that for some of these database, the CC-BY 4.0 license is not applicable. Please find below further details on each of these databanks:

• IBGE (Brazilian Institute of Geography and Statistics) is a federal government institute in Brazil that collects and provides public-domain access to national data. IBGE is the main provider of data and population statistics in the country, serving the needs of the most diverse segments of civil society, as well as the federal, state and municipal government. IBGE offers a complete and current view of the country, through the performance of its main functions, such as: Coordination of Cartographic and Statistics Information Systems, Production of Statistical Information, Production of Geoscientific Information, Production of Census Information, Production of Environmental Information, Dissemination of Information, Higher Education Management, Research and Extension. Of note, all data available at the IBGE website is open to public access. Many papers that used IBGE databases have been published in PLOS ONE over the years, such as: Influence of Environmental Governance on Deforestation in Municipalities of the Brazilian Amazon (https://doi.org/10.1371/journal.pone.0131425); The dynamics of coffee production in Brazil (https://doi.org/10.1371/journal.pone.0219742); Parity of Indigenous and Non-Indigenous Women in Brazil: Does the Reported Number of Children Born Depend upon Who Answers National Census Questions? (https://doi.org/10.1371/journal.pone.0123826). 

• CPRM (Mineral Resources Research Company) is the Brazilian official agency legally bound to gather data and information on Brazilian geology, minerals and water resources. Similarly to IBGE, CPRM is a federal public institute in Brazil that provides public-domain access. CPRM offers a robust set of databases, theme-based georeferenced information systems, documents, maps and images for the general public’s usage. Indeed, many papers that used CPRM databases have also been published in PLOS ONE over the years, such as: Mapping human vulnerability to climate change in the Brazilian Amazon: The construction of a municipal vulnerability index (https://doi.org/10.1371/journal.pone.0190808); Neogene sharks and rays from the Brazilian ‘Blue Amazon’ (https://doi.org/10.1371/journal.pone.0182740); Anthropogenic landscape decreases mosquito biodiversity and drives malaria vector proliferation in the Amazon rainforest (https://doi.org/10.1371/journal.pone.0245087).

• SRTM (Shuttle Radar Topography Mission): We accessed the data from INPE (National Institute for Space Research). SRTM is an international research effort that, through a CC BY 4.0 license, offer free access to data on digital elevation models on a near-global scale to the general public (https://earthobservatory.nasa.gov/features/ShuttleRetrospective/page6.php). Many articles were published in Plos One using data from SRTM platform, such as: Predictive Models of Primary Tropical Forest Structure from Geomorphometric Variables Based on SRTM in the Tapajós Region, Brazilian Amazon (https://journals.plos.org/plosone/article?id=10.1371/journal.pone.0152009); Drivers of metacommunity structure diverge for common and rare Amazonian tree species (https://journals.plos.org/plosone/article?id=10.1371/journal.pone.0188300); Large-Scale Wind Disturbances Promote Tree Diversity in a Central Amazon Forest (https://journals.plos.org/plosone/article?id=10.1371/journal.pone.0103711).

Additionally, we also added in the figures caption the data source and specified that images were used for illustrative purposes only.

REVIEWERS' COMMENTS:

1. Is the manuscript technically sound, and do the data support the conclusions?

Reviewer #1: Yes

Response: No answer needed.

2. Has the statistical analysis been performed appropriately and rigorously?

Reviewer #1: No

Response: Here, we approached the dynamics of plague in the municipality of Exu by using spatial scanning functions to identify spatial and temporal clusters with statistical significance, such as: Retrospective Space-Time analysis scanning for clusters with high rates using the Discrete Poisson model; and Retrospective Space-Time analysis scanning for clusters with short or long survival using the Exponential model. Moreover, we applied the Kernel analysis to identify the location of clusters for case occurrences which is very important to evidence the results here described.

3. Have the authors made all data underlying the findings in their manuscript fully available?

Reviewer #1: No

Response: In this reviewed version we added to the supplemental files our data bank on patients clinical and epidemiological features, as well as a case-by-case sheet with all confirmed plague cases in the State of Pernambuco, including dates, locations and the coordinates for each municipality affected. Furthermore, as mentioned above, we added in the methods session the links to access all the geographical data used in this work. For the analysis performed at the municipality of Exu, we listed the cases by date and by location (up to the name of the districts) and whether it is at the rural or urban zone. However, we cannot publish the precise coordinates of the residences where the cases in Exu took place due to the patient data protection. 

4. Is the manuscript presented in an intelligible fashion and written in standard English?

Reviewer #1: Yes

Response: No answer needed.

5. Review Comments to the Author

Reviewer #1: I feel the paper has the potential to be published in this journal. First, the authors have did a good job to edit the historical data of plague in South America. I appreciate this very much. The data is very valuable with detailed time and location information.

However, the author had a poor data analysis. The paper is too descriptive to be accepted at the current form. There are available approaches to estimate the transmission rates using historical data (e.g. Xu et al. 2014. Wet climate and transportation routes accelerate spread of human plague. Proceedings of the Royal Society B-Biological Sciences. 281: 1780), and then analyze the potential factors affecting the transmission rate, and identify the key factors.

Response: We would like to thank the reviewer for the insightful comments and suggestions. As explained in question #2 our manuscript used spatial statistical analyzes to process data presenting in the results of this study. We believe that combined, these approaches addressed the main objective of the study, which was to better understand the plague dynamics in Pernambuco state, Northeastern Brazil. By analyzing the spatial occurrence and distribution of cases and using the municipality of Exu as study case we were able to characterize the spatial and temporal dispersion of cases. With this in mind, we believe that although the manuscript has some descriptive information, the spatial statistical analyzes support the findings and the whole discussion that has as goal to better understand how the plague occurred in one of the main foci of transmission in Brazil, the Chapada do Araripe in Exu.

We are grateful for the indication of the article “Wet climate and transportation routes accelerate spread of human plague” (Society B-Biological Sciences. 281: 1780) to improve our manuscript, but there are some technical considerations that make such analyzes impossible to be carried out in the territorial unit presented in our study - the county of Exu in Pernambuco. First, the analysis was carried out in a country with continental territory such as China (9,597,000 km²), and our study presents the analysis in a very small territory (Exu: 1,473 km²). In addition, the period studied in China was almost 2 centuries (1772 to 1964) while ours was performed only for 32 years (1945-1976) for the municipality of Exu. These differences compromise data analysis when using methodologies such as Trend-Surface Analysis (TSA) and Nearest Neighbor Approach (NNA). Such methods were used to analyze the plague spread in China considering the location of the cases and the geographic historical data, such as the occurrence of rivers and routes. However, for our territorial unit there are no historical data on roads, which makes it difficult to perform such analysis.

We think that our data analysis approach using Retrospective Space-Time analysis scanning for clusters with short or long survival using the Exponential model clarify how the plague dispersion occurred in our study area (Exu) over the study period.

However, we will consider the methods TSA and NNA to analyze data on the occurrence of plague in Brazil in future studies over larger territories, since this manuscript is part of a larger project that investigate the main foci of the disease in the country. We will be happy to publish them in Plos One as soon as the analyzes are completed.

6. PLOS authors have the option to publish the peer review history of their article (what does this mean?). If published, this will include your full peer review and any attached files.

Do you want your identity to be public for this peer review? For information about this choice, including consent withdrawal, please see our Privacy Policy.

Reviewer #1: No

Response: No answer needed.

---

## [Decision Letter · Decision Letter 1]

19 Mar 2021

Spatiotemporal analysis of bubonic plague in Pernambuco, Northeast of Brazil: case study in the Municipality of Exu.

PONE-D-20-35938R1

Dear Dr. Almeida,

We’re pleased to inform you that your manuscript, "Spatiotemporal analysis of bubonic plague in Pernambuco, Northeast of Brazil: case study in the Municipality of Exu" (PONE-D-20-35938R1), has been judged scientifically suitable for publication and will be formally accepted for publication once it meets all outstanding technical requirements.

Kind regards,

Abdallah M. Samy, PhD

Academic Editor

PLOS ONE

---

## [Editor Report · Acceptance letter]

26 Mar 2021

PONE-D-20-35938R1 

Spatiotemporal analysis of bubonic plague in Pernambuco, Northeast of Brazil: case study in the Municipality of Exu. 

Dear Dr. de Almeida:

I'm pleased to inform you that your manuscript has been deemed suitable for publication in PLOS ONE. Congratulations! Your manuscript is now with our production department. 

Kind regards, 

on behalf of

Dr. Abdallah M. Samy 

Academic Editor

PLOS ONE